# Recovery of Scots Pine Seedlings from Long-Term Zinc Toxicity

**DOI:** 10.3390/plants13162227

**Published:** 2024-08-11

**Authors:** Yury V. Ivanov, Alexandra I. Ivanova, Alexander V. Kartashov, Vladimir V. Kuznetsov

**Affiliations:** K.A. Timiryazev Institute of Plant Physiology, Russian Academy of Sciences, Botanicheskaya Street 35, 127276 Moscow, Russia; aicheremisina@mail.ru (A.I.I.); botanius@yandex.ru (A.V.K.)

**Keywords:** *Pinus sylvestris*, mineral nutrition, manganese, low-molecular-weight antioxidants, phenolic compounds, lignin

## Abstract

We studied the recovery of the growth and physiological parameters of Scots pine seedlings after long-term zinc toxicity. The removal of excess zinc from the nutrient solution resulted in the rapid recovery of primary root growth but did not promote the initiation and growth of lateral roots. The recovery of root growth was accompanied by the rapid uptake of manganese, magnesium, and copper. Despite the maximum rate of manganese uptake by the roots, the manganese content in the needles of the recovering plants did not reach control values during the 28 days of the experiment, unlike magnesium, iron, and copper. In general, the recovery of ion homeostasis eliminated all of the negative effects on the photosynthetic pigment content in the needles. However, these changes, along with recovery of the water content in the needles, were not accompanied by an increase in the weight gain of the recovering seedlings compared with that of the Zn-stressed seedlings. The increased accumulation of phenolic compounds in the needles persisted for a long period after excess zinc was removed from the nutrient solution. The decreased lignin content in the roots and needles is a characteristic feature of Zn-stressed plants. Moreover, the removal of excess zinc from the nutrient solution did not lead to an increase in the lignin content in the organs.

## 1. Introduction

Environmental pollution is currently one of the most serious challenges. In particular, owing to high rates of urbanization and industrialization, soil contamination with toxic heavy metal(loid)s (TMs) has become a worldwide problem [1,2,3]. TMs represent a special group of pollutants because they are not subject to biodegradation or destruction [4]. As a result, TMs can accumulate in soil at extremely high levels, as has been observed in various regions of the world [5,6,7]. This problem is most acute at heavily contaminated industrial production sites that have been abandoned due to high reclamation costs [3,8] and in roadside soil [9,10]. Furthermore, large numbers and areas of heavily contaminated sites imply competition for land use. According to some estimates, there are more than 5 million sites in the world where the soil is contaminated with TMs [3,11].

The areas devoid of vegetation that have arisen near sources of industrial pollution have been classified as industrial barrens. Approximately three-quarters of industrial barrens occur on former forest lands, where they have completely replaced coniferous or mixed forests [12]. Forest dieback in heavily polluted areas occurred even decades after emissions ceased [13]. The forest areas damaged to varying degrees by nonferrous metallurgy are tens of times greater than the area of industrial barrens [14]. In these areas, forests play an important ecological role—phytostabilization of TMs [15]. By stabilizing the soil with roots and reducing soil erosion [16], plants reduce the movement of TMs, preventing the leaching of TMs deeper into the soil profile [1,17].

It is assumed that plants can enter a post-metal stress stage during their life cycle due to various processes that decrease TMs concentrations in the soil [5]. This scenario is more realistic for woody plants owing to their extensive root system and root growth into soil horizons uncontaminated by TMs [18]. Many factors, such as site origin, distance to pollution sources, soil physical and chemical properties, and topography, cause differences in the vertical distribution of TMs in the soil profile [3]. However, significant differences in the vertical distribution of TMs should be expected between highly contaminated sites, such as abandoned smelters or landfills, and sites with moderate and weak contamination, such as roadside soil.

In the area of an abandoned smelter, the lower soil horizons were contaminated with TMs due to their migration from the upper horizons. Cd, Tl, Sb, and As migrated more quickly than Pb and Zn. A high content of Zn was observed in the 1–2 m soil layer [10]. The distributions of Zn, Pb, Cu, Cd, Hg, As, and Mn all tended to decrease with increasing soil depth. The depth of TMs migration was as follows: Cd > Hg > As > Zn > Pb > Cu > Mn [3]. The highest levels of TMs in the soil of a closed unlined landfill were found at depths greater than 2.5 m, and Zn had the highest leaching value of all the TMs studied (Cd, Cr, Cu, Ni, and Pb) [4].

For roadside soil, decreases in the concentrations of Pb, Ni, Cd, and Zn were noted both with distance (up to 32 m) from traffic and with depth (up to 15 cm) of the soil profile. Moreover, the TMs gradient with depth followed the order of Cd > Zn > Pb > Ni [9]. Zn, Pb, Cu, and Cd accumulated in the surface soil, and their concentrations decreased exponentially with increasing soil depth. Moreover, the TMs concentrations in the soil horizons under the TMs accumulation layers were identical to the background values. Evident accumulation of TMs occurred at depths of 20–30 cm at locations of 50 m from roads [10]. Moreover, soil under forests was characterized by a lower accumulation of TMs than soil in other land use types [10].

Previously, we showed that the long-term impact of cement plant emissions leads to a significant accumulation of TMs (Pb, Cu, Ni, and Zn) in the upper horizons of forest soil at depths of up to 15 cm. At the same time, at a depth of 15–25 cm, the Cu and Pb contents were tens of times lower, and the Zn and Ni contents were several times lower than those in the 5–15 cm horizon [18]. Thus, different contaminants have various migration depths, and even in heavily polluted areas, lower soil layers with background values of TMs are potentially available for the roots of adult trees [19]. Therefore, if plant roots are able to cross the metal-contaminated soil horizon, plant growth and development can recover.

Scots pine (*Pinus sylvestris* L.) is an important tree species in Northern Eurasia [20] that is sensitive to TMs [13,21,22]. For example, Scots pine is very sensitive to the long-term toxicity of Zn or Cu from the seed germination stage [14,23], resulting in strong root growth inhibition, which can prevent forest renewal on contaminated soils [6]. Moreover, short-term exposure of Scots pine seedlings with a well-developed root system to excess Zn or Cu led to mineral nutrition disorders and a deficiency of phenolic compounds, which was especially pronounced under Zn toxicity [24]. However, there are currently no data on the recovery of Scots pine from Zn toxicity.

The objective of this study was to test the hypothesis that the removal of excess Zn from nutrient solution is accompanied by the restoration of root system growth in Scots pine seedlings and an increase in their capacity to uptake essentials nutrients, resulting in the normalization of ion homeostasis and restoration of physiological functions. To test this hypothesis, we studied the morphological and physiological parameters, balance of essential elements, and dynamics of low-molecular-weight antioxidants and lignin contents in the organs of the seedlings throughout 28 days of recovery from long-term Zn stress.

## 2. Results

### 2.1. Development of Scots Pine Seedlings at the Beginning of the Experiment

At the beginning of the experiment, compared with control plants, Scots pine plants grown under long-term exposure to zinc (Zn) (46 days after massive seed germination) were characterized by a significantly greater Zn content in their organs (Table 1). Thus, the Zn content in the roots exceeded that in the roots of the control by 21.7 times, that in the hypocotyls by 8.4 times, that in the cotyledons by 6.5 times, and that in the needles by 8.7 times. Moreover, the fresh weight of the pine seedlings (257.5 ± 9.1 mg) grown under long-term exposure to Zn was 23.7% lower than that of the control plants (337.5 ± 13.3 mg) (Figure 1a). The fresh weights of the roots and needles were 27.0% and 26.8% lower than those of the control, respectively (Table 1). The dry weight of the plants was 18.5% lower than that of the control plants (Figure 1b) because of the lower dry weights of the needles (30.2%) and roots (27.0%) (Table 1). The water content in whole seedlings under Zn stress was 1.0% lower than that in the control (Figure 1c), which was due mainly to the reduced water content in the needles (Table 1). The long-term toxic effects of Zn clearly manifested in the inhibition of the development of seedling organs: the length of the primary root was 44.7% shorter than that in the control (Table 1, Figure 2a). The lengths of the hypocotyls and cotyledons were 6.2% and 8.3%, respectively, less than those in the control. The diameter of the hypocotyl (Table 1, Figure 3a), the length of the epicotyl, and the number of needles (Figure 3b,c) were 11.0%, 28.4%, and 10.7%, respectively, less than those in the control.

### 2.2. Seedlings Growth during Recovery from Zn Toxicity

During the 28-day experiment, the fresh weight of the control seedlings increased by 1.8 times, the fresh weight of the Zn-stressed seedlings increased by 1.9 times, and the fresh weight of the plants that recovered from Zn toxicity increased by 2.2 times. Owing to the high variability of the seedling fresh weights at the end of the experiment, there were no significant differences between the experimental groups. During the experiment, the dry weight of the plants increased 3.7-fold in the control group, 3.3-fold under Zn toxicity, and 3.2-fold during recovery from Zn toxicity (Figure 1a,b). Two-way ANOVA of the data on the fresh and dry weights of the seedlings revealed significant differences between the experimental groups throughout the experiment. However, it should be noted that the removal of excess Zn from the nutrient solution was not accompanied by an increase in the weight gain of the recovering seedlings compared with that of the Zn-stressed seedlings.

After the removal of excess Zn from the nutrient solution, the water content in the recovering seedlings no longer differed from that in the control plants, in contrast to that in the Zn-stressed seedlings. A clear trend toward an increase in the water content in the recovering seedlings compared with that in the Zn-stressed plants was observed starting from the 5th day of the experiment. However, a statistically significant difference between these variants was noted only for a single time point (10th day). A characteristic feature of all of the experimental groups was a significant decrease in the water content in the seedlings from the 21st to 28th days of the experiment (Figure 1c).

Two-way ANOVA of the fresh and dry weights of the seedling roots indicated that the effect of time dominated the effect of the experimental groups (Appendix A). Thus, the increase in the fresh weight of the roots during the experiment was more pronounced for the recovering seedlings: 121.0% in fresh weight and 3.5 times in dry weight, respectively, versus 95.1% and 3.2 times in the Zn-stressed seedlings and 72.5% and 3.1 times in the control, respectively (Table 2). For the fresh and dry weights of the needles, significant influences of both factors (experimental variant and time) were clearly noted (Appendix A). The increase in the fresh weight of the needles at the end of the experiment compared with that at the initial point was more pronounced in the recovering seedlings (2.8 times) than in the control (2.5 times) and Zn-stressed seedlings (2.4 times). At the end of the experiment, the fresh weight of the needles of the control seedlings was 27.7% greater than that of the Zn-stressed seedlings. Moreover, the increase in the dry weight of the needles was more pronounced in the control variant (3.7 times), versus 3.5 times in the Zn-stressed seedlings and 3.4 times in the recovering seedlings (Table 2).

Similar to the results for the needles, 2-way ANOVA of hypocotyl fresh and dry weights demonstrated the same results (Appendix A). The increase in the fresh weight of the hypocotyls was more pronounced for the recovering seedlings (2.3 times greater during the experiment), for the control (2.2 times greater), and for the Zn-stressed seedlings (2.1 times greater). A considerable increase in the dry weight of hypocotyls was noted in the control—3.1 times, recovering seedlings—2.6 times, and Zn-stressed plants—2.5 times (Table 2). Neither the fresh nor the dry weights of the cotyledons changed during the experiment (Table 2), which was confirmed by 2-way ANOVA, indicating the absence of a significant influence of time (Appendix A).

The water content in the roots of the recovering plants increased significantly compared with that in the roots of the Zn-stressed plants on the 7th and 10th days of the experiment. However, 2-way ANOVA did not reveal any relationship with the experimental variants, only with sampling time (Appendix A). This was probably due to a pronounced decrease in the water content in the roots of all the experimental variants from the 21st to the 28th day of the experiment. Compared with that at the initial point, the water content in the roots on the 28th day did not change significantly in any of the experimental variants (Table 2).

No significant changes in the water content in the hypocotyls were noted either between experimental variants or with sampling time. In the cotyledons, 2-way ANOVA revealed clear differences between the experimental groups, but pairwise *t*-test comparisons of the Zn-stressed and recovering plants revealed significant differences at only a single time point (7th day). This indicated the absence of pronounced dehydration of the cotyledons during the experiment, including Zn-stressed plants (Table 2, Appendix A).

In the needles, there was a clear trend of decreasing water content throughout the experiment, which was typical for all of the experimental variants. However, under Zn toxicity, this decrease in water content was more pronounced and accelerated starting on the 17th day of the experiment. The water content in the needles of the recovering plants recovered to a level comparable to that in the needles of the control plants beginning on the 5th day, although it decreased on the 10th and 14th days of the experiment. Despite the obvious increase in the water content in the needles of the recovering plants, significant differences from that in the needles of the Zn-stressed plants were observed only starting from the 21st day of the experiment (Table 2, Appendix A).

For the data characterizing the development of the seedling root system (Figure 2), 2-way ANOVA did not allow us to properly interpret the effects of variant or time because the size of the factor’s effect depended upon the level of the other factor.

The length of the primary root of the control plants remained virtually unchanged throughout the experiment. Moreover, the length of the primary root of the Zn-stressed plants increased by 48.8% by the end of the experiment. The length of the primary root of the recovering seedlings increased by 81.0% during the experiment, and on the 28th day, this value did not significantly differ from that of the control (Figure 2a).

Significant differences in the distance from the tip of the primary root to the first lateral root between the Zn-stressed and recovering plants were noted beginning on the 1st day of the experiment. In the control, despite changes in this parameter during the experiment, the initial and end (28th day) points did not differ. The distance from the tip of the primary root to the first lateral root of the Zn-stressed plants decreased by 26.0% at the end of the experiment, whereas for the recovering plants, the distance increased by 109.4% (Figure 2c).

The increase in the number of first-order lateral roots continued in all of the experimental groups throughout the experiment. In the control group, this increase was 46.9%; in the Zn-stressed group, it was 81.8%; and in the recovering group, it was 91.4%. Moreover, at most time points, the number of first-order lateral roots in the control seedlings exceeded their number in the other experimental groups. Thus, the removal of excess Zn from the nutrient solution did not lead to a clear increase in the number of first-order lateral roots in the recovering seedlings (Figure 2b).

The number of second-order lateral roots increased by 4.4 times in the control, by 2.8 times in the Zn-stressed seedlings, and by 4.1 times in the recovering seedlings during the experiment. Generally, pairwise comparisons of the means at each time point did not reveal statistically significant differences between the control and experimental groups in most cases. As with the first-order lateral roots, the removal of excess Zn from the nutrient solution did not lead to an obvious increase in the number of second-order lateral roots (Figure 2d).

During the experiment, the hypocotyl diameter increased by 20.9% in the control plants, 27.2% in the Zn-stressed plants, and 29.6% in the recovering plants. The significant differences between the hypocotyl diameters of the control and Zn-stressed seedlings were maintained throughout the experiment, whereas the significant differences between the recovering plants disappeared starting on the 17th day of the experiment. Moreover, significant differences between the Zn-stressed and recovering plants were noted only on the 5th day of the experiment, i.e., they were of a random nature (Figure 3a).

At the beginning of the experiment, the epicotyl length of the control plants exceeded the epicotyl length of the Zn-stressed plants by 28.4%. During the experiment, the epicotyl length increased by 3.1-fold for both the control and the Zn-stressed plants and by 3.4-fold for the recovering plants. As a result, by the end of the experiment, the differences between the experimental variants were comparable to the differences at the initial point. No significant differences were detected between the Zn-stressed and recovering plants (Figure 3b).

At the beginning of the experiment, the Zn-stressed seedlings were characterized by 10.7% fewer needles than in the control. During the experiment, the number of needles increased by 94.5% in the control, 86.7% in the Zn-stressed plants, and 96.2% in the recovering plants. Two-way ANOVA revealed significant influences of both factors (experimental variant and time). However, statistically significant differences between the Zn-stressed and recovering seedlings were detected only on the 5th day of the experiment. Pairwise comparisons of the number of needles in recovering seedlings with that in the control revealed non-significant differences at most time points. This indicated an intensification of the formation of new needles after the removal of excess Zn from the nutrient solution (Figure 3c).

### 2.3. Ion Homeostasis during Recovery from Zn Toxicity

#### 2.3.1. Zinc

The Zn content in the roots of the control plants remained almost constant throughout the experiment. In the Zn-stressed plants, slight fluctuations in the Zn content in the roots were observed throughout the experiment. However, at the end of the experiment, the Zn content in the roots did not differ from that at the initial point. The Zn content in the roots of the recovering plants decreased significantly beginning on the 3rd day of the experiment and was 30.2% lower than that in the roots of the Zn-stressed plants. A progressive decrease in the Zn content in the roots of the recovering seedlings was observed until the end of the experiment, with a clear trend from the 3rd to 17th days of the experiment. As a result, at the end of the experiment, the Zn content in the roots of the recovering plants was only 3.1 times greater than that in the roots of the control, whereas the Zn content in the roots of the Zn-stressed plants exceeded that in the roots of the control by 14.3 times (Figure 4a, Appendix A).

The Zn content in the hypocotyls of the recovering plants decreased starting on the 7th day of the experiment. At the end of the experiment, the Zn content in the hypocotyls of the recovering plants was 4.3 times greater than that in the hypocotyls of the control, whereas in the hypocotyls of the Zn-stressed plants, it was 11.3 times greater than that in the hypocotyls of the control (Appendix A).

For the seedling cotyledons, no significant changes in Zn content were noted throughout the experiment. The removal of excess Zn from the nutrient solution did not lead to changes in the Zn content in the cotyledons (Appendix A).

The Zn content in the needles of the control and Zn-stressed plants remained stable throughout the experiment (Figure 4b, Appendix A). The Zn content in the needles of the recovering plants decreased starting on the 14th day of the experiment. As a result, at the end of the experiment, the Zn content in the needles of the recovering plants was 3.6 times greater than that in the needles of the control, whereas in the Zn-stressed plants, it was 7.6 times greater than that in the needles of the control.

Recalculation of the Zn content data into the Zn amount per organ indicated that during recovery, the Zn amount in the roots sharply decreased during the first 3 days after removal of excess Zn from the nutrient solution (by 32.4% from the initial point) and then decreased by the 7th day of the experiment (by 44.2% from the initial point). After that, the amount of Zn in the roots remained almost constant until the end of the experiment (Appendix A). An increase in the amount of Zn in all of the organs of the Zn-stressed plants was observed during the experiment. Significant differences in the amount of Zn in the needles of the recovering and Zn-stressed plants were observed only beginning on the 17th day of the experiment (Appendix A).

#### 2.3.2. Magnesium

At the beginning of the experiment, the Mg content in the roots and needles of the Zn-stressed plants decreased by 37.4% and 26.2%, respectively, compared with that in the control plants (Figure 4c,d). In the hypocotyls and cotyledons, no significant changes in Mg content were detected between the experimental groups at the beginning of the experiment (Appendix A).

The Mg content in the roots of the recovering plants clearly exceeded that in the roots of the Zn-stressed plants starting from the 7th day of the experiment, and on the 17th day, it reached the control level. The difference between the control and the Zn-stressed plants at the end of the experiment was 41.6%. The active uptake of Mg into the roots after the removal of excess Zn from the nutrient solution was evidenced by changes in the amount of Mg, which significantly differed from that in the Zn-stressed plants beginning on the 5th day of the experiment (Appendix A).

The Mg content in the hypocotyls of the Zn-stressed plants decreased starting on the 17th day of the experiment, whereas the Mg content in the hypocotyls of the recovering plants was comparable to that in the control plants throughout the experiment. For the cotyledons, no pronounced changes in the Mg content were observed between the experimental variants throughout the experiment (Appendix A).

In the needles of the recovering seedlings, the Mg content began to differ from that in needles of the Zn-stressed plants only starting from the 10th day of the experiment, and on the 17th day, it reached the level of that in the needles of the control plants (Figure 4d).

#### 2.3.3. Iron

Long-term Zn toxicity did not affect the Fe content in the organs of the seedlings at the beginning of the experiment (Figure 4e,f, Appendix A). Generally, the Fe content in the roots tended to increase throughout the experiment. Despite the significant influence of the experimental variants (Appendix A), no significant changes were observed between the recovering and control seedlings. Nevertheless, the Fe content in the roots of the recovering plants was greater than that in the roots of the Zn-stressed plants on the 17th and 21st days of the experiment (Figure 4e).

The Fe content in the needles of the control plants was stable throughout the experiment. Interestingly, the excess Fe content in the needles of the recovering plants was greater than that in the needles of the control plants starting on the 5th day of the experiment. As a result, at the end of the experiment, the Fe content in the needles of the recovering plants was 21.3% greater than that in the needles of the control plants (Figure 4f). Greater amounts of Fe were detected in the needles of the recovering plants than in those of the Zn-stressed plants starting from the 5th day of the experiment at all points except for the 14th day. For the hypocotyls and cotyledons, high variability in the Fe content was observed throughout the experiments (Appendix A).

#### 2.3.4. Manganese

At the beginning of the experiment, the Mn contents in the organs of the Zn-stressed plants were significantly lower than those in organs of the control plants; by 4.2 times in the roots, by 47.8% in the needles (Figure 4g,h), and by 20.4% in the cotyledons. There were no differences in the Mn content in the hypocotyls (Appendix A). In the Zn-stressed seedlings, the development of Mn deficiency progressed throughout the experiment and decreased by 67.0% in the roots, 32.9% in the hypocotyls, and 24.6% in the needles at the end of the experiment compared with that at the initial point.

The removal of excess Zn from the nutrient solution led to a significant increase in the Mn content in the roots of the recovering seedlings within one day. Throughout the experiment, the Mn content in the roots of the recovering plants was greater than that in the roots of the Zn-stressed plants. Due to the high variability of the Mn content in the roots of the control plants, the Mn content in the roots of the recovering plants reached values comparable to that in the roots of the control plants after 3–7 days of the experiment (Figure 4g). The amount of Mn in the roots of the recovering seedlings reached a plateau on the 5th day of the experiment and did not increase thereafter (Appendix A). A significant increase in the Mn content in the needles of the recovering seedlings was noted starting from the 5th day of the experiment. At the end of the experiment, the Mn contents in the roots and needles of the recovering plants were 2.7 times and 31.5%, respectively, lower than those in the roots and needles of the control (Figure 4h). The Mn contents in the roots and needles of the Zn-stressed plants were 12.1 times and 64.2%, respectively, lower than those in the roots and needles of the control plants.

#### 2.3.5. Copper

The long-term toxicity of Zn did not affect the Cu content in the organs of the seedlings at the beginning of the experiment (Figure 4i,j, Appendix A). The Cu content in the roots of the control plants did not change throughout the experiment. Moreover, the Cu content in the roots of the Zn-stressed seedlings increased by 17.9% at the end of the experiment compared with that at the initial point, whereas in the recovering plants, it increased by 42.8%. Notably, the removal of excess Zn from the nutrient solution resulted in an increase in the Cu content in the roots of the recovering seedlings compared with that in the roots of the control, which was most pronounced toward the end of the experiment (Figure 4i).

High variability in the Cu content in both the hypocotyls and cotyledons was observed throughout the experiment (Appendix A). Two-way ANOVA of the Cu content revealed a significant interaction effect of both factors (experimental variant and time) on both organs (Appendix A). Generally, in the hypocotyls of the Zn-stressed plants, there was a trend toward a lower Cu content, whereas in the cotyledons, a greater Cu content was observed. However, the removal of excess Zn from the nutrient solution led to an increase (at several time points) in the Cu content in the hypocotyls but did not affect the cotyledons (Appendix A).

Two-way ANOVA of the Cu content in the needles revealed significant influences of both factors (experimental variant and time) (Appendix A). Generally, Zn toxicity resulted in a decreased Cu content in the needles. The removal of excess Zn was accompanied by a recovery of the Cu content in the needles to control levels. As a result, at the end of the experiment, the Cu content in the needles of the recovering plants did not differ from that in the needles of the control, whereas in the Zn-stressed plants, it was 14.7% lower than that in the needles of the control (Figure 4j).

### 2.4. Photosynthetic Pigments

The content of photosynthetic pigments in the control seedlings differed significantly between the top needles and basal needles. In particular, the contents of chlorophylls *a* and *b* and carotenoids in the top needles were 25.7%, 24.7%, and 24.4%, respectively, lower than those in the basal needles (Table 3).

At the beginning of the experiment, the basal needles of the Zn-stressed seedlings were characterized by a lower content of photosynthetic pigments than that in the basal needles of the control plants. In particular, the contents of chlorophylls *a* and *b* in the basal needles were 7.3% and 8.0% lower, respectively, than those in the basal needles of the control. The content of carotenoids was 6.9% lower than in the control. Moreover, no differences were noted in the ratio of chlorophylls or the ratio of carotenoids to chlorophylls (Table 3). In the top needles, significant differences were found only for the content of carotenoids, which was 7.1% lower in the Zn-stressed plants than in the control plants.

At the end of the experiment, the basal needles were characterized by an increase in the content of photosynthetic pigments (Table 3, Appendix A). Although there were no differences between the recovering and control seedlings, the pigment contents in the basal needles of the Zn-stressed plants were 14.4%, 16.4%, and 13.2% lower than those in the basal needles of the control plants for chlorophylls *a* and *b* and carotenoids, respectively. In addition, a characteristic feature was a decrease in the ratios of chlorophylls *a* and *b* and carotenoids to total chlorophylls at the end of the experiment compared with those at the initial point.

The content of photosynthetic pigments in the top needles of the seedlings did not exhibit a pronounced time trend (Appendix A). At the end of the experiment, no differences in pigment content were observed between the control and recovering seedlings. However, there was a significantly greater content of photosynthetic pigments in the top needles of the recovering plants than in the top needles of the Zn-stressed plants. Like those in the basal needles, the ratios of chlorophylls *a* and *b* and carotenoids to total chlorophylls decreased in the upper needles at the end of the experiment (Table 3).

### 2.5. Contents of Low-Molecular-Weight Antioxidants, Total Phenolics, Catechins, and Proanthocyanidins during Recovery from Zn Toxicity

At the beginning of the experiment, no significant differences in the TEAC were detected between the roots and needles of the seedlings (Figure 5). However, the roots of the Zn-stressed plants were characterized by a lower TEAC than in the roots of the control plants throughout the experiment. By contrast, the needles of the Zn-stressed seedlings had an increased TEAC (Figure 5, Appendix A). At the end of the experiment, the TEAC in the roots of the Zn-stressed plants was 23.6% lower, whereas that in the needles was 58.1% greater than that in the roots of the control (Figure 5). The TEAC of the hypocotyls decreased during the experiment, which was confirmed by 2-way ANOVA (Appendix A). Moreover, compared with those of the control plants, the hypocotyls and cotyledons of the Zn-stressed plants were characterized by increased TEAC values, especially toward the end of the experiment. However, the TEAC in hypocotyls and cotyledons tended to decrease throughout the experiment (Appendix A). Despite the observed increase in the TEAC in the roots of the recovering plants starting on the 10th day of the experiment, this increase did not have a stable trend and did not differ significantly from that in the roots of the Zn-stressed plants (Figure 5, Appendix A). The needles of the control plants tended to exhibit a decrease in the TEAC throughout the experiment. As a result, on the 28th day of the experiment, the TEAC level was 20.2% lower than that at the initial point. By contrast, the TEAC in the needles of the Zn-stressed plants increased significantly and was 58.1% greater than that in the needles of the control plants. In the recovering plants, the TEAC in the needles tended to decrease from the 17th day to the 28th day of the experiment, and on the 28th day, it exceeded that in the needles of the control by only 18.2% (Figure 5).

A pattern similar to that of the TEAC content was characteristic of the GAE content in the organs of the seedlings. The exception was the roots of the control and recovering plants, the GAE content of which tended to decrease from the 21st to the 28th days of the experiment. Significant changes in the roots of the Zn-stressed plants compared with those of the control plants were observed only from the 5th to 10th days of the experiment (Appendix A). Compared with those of the control plants, the needles of the Zn-stressed plants were characterized by an increase in the content of GAEs throughout almost the entire experiment. A decrease in the GAE content in the needles of the recovering plants compared to that in the needles of the Zn-stressed plants was noted only on the 28th day of the experiment. As a result, the GAE content in the needles of the Zn-stressed plants exceeded that in the needles of the control plants by 46.8%, and that in the needles of the recovering plants exceeded that in the needles of the control plants by only 19.3% (Figure 5, Appendix A).

The total catechins and PAs content in the roots and needles generally reflected the dynamics of the TEAC and GAE during the experiment. However, due to the high variability, significant differences in the content of catechins and PAs in the roots were revealed in the middle of the experiment (7th and 10th days), in contrast to the dynamics of the TEAC (Appendix A). Two-way ANOVA of the catechins and PAs content in the needles revealed significant influences of both factors (experimental variant and time) (Appendix A). In general, the needles of the Zn-stressed seedlings presented increased levels of catechins and PAs throughout the experiment, whereas the needles of the recovering plants tended to have a decreased catechins and PAs content. Moreover, statistically significant differences between the Zn-stressed and recovering seedlings were detected starting on the 5th day of the experiment (Appendix A). As a result, at the end of the experiment, the total catechins and PAs content in the needles of the recovering plants did not differ from that in the needles of the control, whereas in the needles of the Zn-stressed plants, the total catechins and PAs content was 77.1% greater than that in the needles of the control (Figure 5).

At the beginning of the experiment, the PAs content in the roots of the Zn-stressed seedlings was 18.3% lower than that in the roots of the control plants. At the end of the experiment, these differences increased to 34.2% (Figure 5). Two-way ANOVA revealed significant influences of the experimental variants (Appendix A). This was explained by the fact that, at all studied time points, the PAs content in the roots of the Zn-stressed plants was lower than that in the roots of the control plants. Moreover, in the roots of the recovering seedlings, no significant differences from those of the control plants were noted starting from the 10th day of the experiment. Despite the fact that at the beginning of the experiment there were no significant differences in the PAs content in the needles, there was a clear trend toward an increase in the PAs content in the needles of the Zn-stressed plants during the experiment. As a result, on the 28th day of the experiment, the PAs content in the needles of the Zn-stressed plants was 2.7 times greater than that in the needles of the control plants. Moreover, the PAs content in the needles of the recovering plants was comparable to that in the needles of the control plants (Figure 5).

### 2.6. Lignin Content during Recovery from Zn Toxicity

The lignin content in the roots of the Zn-stressed seedlings at the beginning of the experiment was 23.5% lower than that in the roots of the control plants (Figure 5). Two-way ANOVA of the lignin content in the roots revealed significant influences of both factors (experimental variant and time) (Appendix A). Thus, throughout the experiment, the lignin content increased by an average of 36.9% in the roots of the plants in all of the experimental groups. Moreover, the removal of excess Zn from the nutrient solution did not lead to an increase in the lignin content in the roots of the recovering plants compared with that in the roots of the Zn-stressed plants. At the end of the experiment, the lignin content in the roots of the control plants exceeded that in the roots of the Zn-stressed plants by 17.9%. At the beginning of the experiment, we detected no differences in the lignin content between the control and Zn-stressed plants. However, during the experiment, the rate of lignification of the needles of the control plants was significantly greater than that of the Zn-stressed and recovering plants. Thus, on the 28th day of the experiment, the lignin content in the needles increased by 28.2%, 14.3%, and 12.0% compared with that in the needles at the initial point for the control, Zn-stressed, and recovering seedlings, respectively. As a consequence, at the end of the experiment, significant differences appeared between the control and Zn-stressed seedlings. There were no changes in the lignin content in the needles of the recovering plants compared with that in the needles of the Zn-stressed plants (Figure 5).

Analysis of the lignin content in the epicotyls (after removing the needles) of the plants on the 28th day of the experiment did not reveal any differences between the experimental groups (Figure 6).

## 3. Discussion

### 3.1. Restoration of Growth Processes

Suppression of primary root elongation and impaired growth of lateral roots are considered the main consequences of TMs toxicity in plants [25]. In natural environments, the development of the root system is ensured through the development of separate sections of the roots or even individual roots. Therefore, root avoidance—the plant’s ability to stop root growth or grow away from contaminated sites—is considered to be one of the protective mechanisms against TMs toxicity [2]. Scots pine is known to be a tree species with an “exploitative” root growth pattern that results in greater root length in nutrient-rich soil [26]. Previously, we showed that Zn toxicity in Scots pine seedlings resulted in changes in root architecture and growth inhibition due to the cytotoxic effects of Zn [14]. Although the experimental system we used did not allow for the creation of a Zn toxicity gradient for different zones of the root system, one could expect a clear development of recovery processes after excess Zn removal from the nutrient solution for the entire root system as a whole.

Elimination of the long-term toxic effect of Zn was accompanied by rapid recovery of the growth of the primary root in length (at an average rate of 5.0 mm per day) due to both an increase in the length of the zone from the tip of the primary root to the first lateral root and the zone of lateral roots (Figure 2). Moreover, despite the lengthening of the lateral root zone, increases in the numbers of first-order and second-order lateral roots were not observed. Thus, we can conclude that the elimination of Zn toxicity was accompanied by rapid recovery of root growth but did not promote lateral root initiation and emergence from the parental roots.

In contrast to the rapid recovery of primary root growth, the elimination of Zn toxicity did not result in increased growth of the epicotyl (Figure 3b). However, an intensification of new needle formation in the recovering plants (Figure 3c) indicated the restoration of shoot apical meristem activity after the removal of excess Zn from the nutrient solution.

TMs stress is frequently accompanied by water deficit [5,21]. In many cases, water uptake by plants is indirectly regulated by changes in root anatomy and/or morphology [25]. Thus, the observed suppression of root growth in the Zn-stressed seedlings could lead to a decrease in the root water content. However, no such effect was found throughout the experiment (Table 2). A pronounced decrease in the water content in the roots of Zn-stressed Scots pine seedlings occurred only at higher (300 μM) Zn concentrations, which had a stronger inhibitory effect on root growth [24]. However, the decreased water content in the needles of the Zn-stressed plants (Table 2) was also noted earlier with both long-term [14] and short-term [24] exposure to the same concentration of Zn (150 μM), which did not cause a decrease in the water content in the roots. A possible explanation for this phenomenon may be that TMs-induced vacuolation in root cells and an increase in the volume of vacuoles [25] also develop in response to Zn exposure [27]. However, this mechanism appears to be limited to cells that are not severely damaged by metal ions [25]. Therefore, extensive tissue damage may cause root dehydration, which was observed at higher Zn concentrations [24]. However, despite maintaining the water content in the roots of the Zn-stressed seedlings, progressive dehydration of the needles was observed throughout the experiment (Table 2), probably due to the inhibition of long-distance water transport [25,28]. Moreover, the removal of excess Zn from the nutrient solution was able to stop dehydration and restore the water content in the needles of recovering seedlings.

### 3.2. Recovery of Ion Homeostasis

Surprisingly, despite the recovery of root growth, the recovery of water content in the needles, and the formation of new needles in the recovering seedlings, this recovery was not accompanied by an increase in weight gain of the recovering seedlings compared with the Zn-stressed seedlings (Figure 1a,b). This may be due to severely delayed Zn toxicity, which was previously noted in experiments with *Lemna minor* L. [29]. In particular, the Zn-exposed plants did not recover even 7 days after cessation of Zn exposure, which was in sharp contrast to the results for the other metals (Cu, Ni, and Cs), for which significant recoveries were observed [29]. As shown in Figure 4a, the Zn content in the roots of the recovering seedlings quickly decreased during the 3 days of the experiment. The amount of Zn in the roots remained virtually unchanged after the 7th day of the experiment (Appendix A), despite the increase in the dry weight of the roots (Table 2). In general, only two processes are responsible for the decrease in the Zn content in the roots: leakage of Zn from the roots and “dilution” due to an increase in root weight. A high leakage rate of Zn from the roots of *Salix viminalis* L. was observed at the beginning of the recovery period, indicating the leakage of Zn from the free space compartment, while it was essentially lower at the end of the recovery period [30]. Thus, we can conclude that the decrease in the Zn content in the roots of the Scots pine seedlings throughout the experiment could be due to both the leakage of Zn from the roots, especially at the beginning of the experiment, and its transport to aboveground organs. The latter statement is supported by the fact that a decrease in the Zn content in the needles is possible only because of an increase in needle weight. However, a decrease in the Zn content in the needles of the recovering plants in comparison with that at the initial point occurred only on the 14th day of the experiment (Figure 4b), despite a pronounced increase in the dry weight of the needles throughout this period (Table 2).

Excess Zn is known to disturb certain physiological processes by competing with other ions (Mg, Fe, Mn, and Cu) for binding at absorption regions or loading regions of roots. Moreover, Zn not only interferes with nutrient uptake but also hinders nutrient allocation to different parts of plants [28]. Previously, we demonstrated that excess Zn leads to significant decreases in the contents of Mn and Mg in the organs of Scots pine [14] and, to a lesser extent, in the contents of Fe and Cu [24]. A similar pattern in the contents of the abovementioned ions in the organs of the Zn-stressed seedlings was observed at the beginning of the experiment (Figure 4). Thus, it could be expected that the recovery of root growth after the removal of excess Zn from the nutrient solution (Figure 2) would be accompanied by the recovery of plant ionic homeostasis.

The Mn content in the organs of the seedlings was most affected by Zn toxicity, and unlike other studied ions, the Mn content decreased in the roots and needles of the Zn-stressed plants throughout the experiment. Moreover, the Mn content in the roots and needles of the recovering plants was restored at the fastest rate among the studied ions (Mg, Fe, and Cu). However, despite these findings, the Mn content in the needles of the recovering plants did not reach control levels during the experiment. This significantly distinguished Mn from the other studied ions, the contents of which in the roots and needles of the recovering plants reached or even exceeded the control levels (in the cases of Fe in needles and Cu in roots) (Figure 4). Due to the low mobility of Mn in plants [31], we did not observe remobilization of Mn from Mn-rich cotyledons even under progressive Mn deficiency in the roots and needles of Zn-stressed seedlings. A similar statement is true regarding the lack of remobilization of Mg from the cotyledons (Appendix A).

### 3.3. Photosynthetic Pigments

Disturbed mineral nutrition (Fe, Mn, and Mg) and subsequent changes in the chlorophyll content are considered to be the main effects of Zn toxicity on photosynthesis [2,28]. However, the observed decrease in the content of photosynthetic pigments in the Zn-stressed seedlings was weaker and more pronounced in the basal needles than in the top needles (Table 3). Moreover, the typical symptoms of chlorosis due to deficiency of essential elements first develop in younger leaves (top needles) [31,32]. We can exclude Fe deficiency as a possible cause of chlorosis since the Fe content in the needles of the Zn-stressed seedlings was comparable to that in the needles of the control seedlings (Figure 4f). Thus, Scots pine is characterized as a species that retains a high level of Fe in its needles under Zn stress [28]. On the other hand, Mn deficiency was also not the cause of the decreased pigment content in the needles of the Zn-stressed plants, since even an 8.8 times lower Mn content in the needles than that observed at the beginning of the experiment (Figure 4h) did not lead to a decrease in the content of photosynthetic pigments [32]. Thus, Mg deficiency induced by Zn toxicity (Figure 4d) can be considered the most likely cause of the decreased content of photosynthetic pigments in the needles of the Zn-stressed plants. Generally, the recovery of ion homeostasis after the removal of excess Zn from the nutrient solution eliminated all of the negative effects on the photosynthetic pigment contents in the needles of the recovering seedlings.

### 3.4. Phenolic Compounds and Lignin Content

The recovery of plant growth is accompanied by the restoration of physiological and biochemical processes, at least to some extent [5]. We previously showed that Zn toxicity to Scots pine seedlings led to a pronounced decrease in the TEAC, GAE, PAs, and catechins contents in the roots. Moreover, this effect was specific for Zn (but not for Cu), depended on the concentration of Zn in the nutrient solution, and did not lead to changes in the contents of these substances in the needles of the seedlings [24]. In contrast, long-term exposure to Zn was accompanied by an increase in the content of phenolic compounds in the needles of seedlings [14]. Decreases in the TEAC and phenolic compounds contents in the roots of Zn-stressed plants with a simultaneous increase in the phenolic compounds content in the needles was also a characteristic feature of the present study (Figure 5). Moreover, the similar dynamics of the contents of GAE, catechins, and PAs with the dynamics of TEAC in the organs of the seedlings indicate that the principal composition of the low-molecular-weight antioxidants in these organs did not change during the experiment.

Previously, we suggested that the decrease in the content of phenolic compounds in the roots of Scots pine seedlings under short-term Zn toxicity was due to inhibited synthesis of phenolic compounds because of Mn deficiency in the roots [24]. Notably, the Mn content in the organs was most affected by Zn toxicity. However, the recovery rate of the Mn content in the organs of the seedlings after excess Zn removal was the highest among the studied ions (Figure 4). On the 3rd day of the experiment, its content in the roots of the recovering plants was comparable to that in the roots of the control plants. However, the increase in the TEAC in the roots of the recovering plants compared with that in the roots of the Zn-stressed plants occurred only from the 10th day of the experiment (Figure 5), whereas the restoration of the contents of GAE, catechins, and PAs to control levels was noted only from the 14th day of the experiment (Appendix A). Moreover, on the 7th day of the experiment, the Mg content in the roots of the recovering plants increased substantially and was only 18.1% lower than that in the roots of the control (Figure 4c). This fact does not make it possible to associate the increase in the content of phenolic compounds in the roots of recovering plants during the experiment only with the recovery of the Mn content in the roots. Therefore, this assumption requires additional research. Moreover, the GAE contents in the roots of the Zn-stressed and recovering seedlings did not differ from each other at the end of the experiment, i.e., these changes are unlikely to be related to the Mn or Mg contents in the roots.

In contrast to those of the roots, the needles of the Zn-stressed seedlings were characterized by an increase in the TEAC and the contents of the studied low-molecular-weight antioxidants throughout the experiment. A pronounced decrease in the content of phenolic compounds in the needles of the recovering seedlings compared with that in the needles of the Zn-stressed seedlings occurred only at the end of the experiment (Figure 5). Considering the twofold increase in the number of needles during the experiment (Figure 3c) and the almost threefold increase in the fresh weight of the needles of the recovering seedlings (Table 2), it can be argued that the increased accumulation of phenolic compounds in the needles persisted for a long period after excess Zn was removed from the nutrient solution.

An increase in lignin content in response to excess metals (Cu, Zn, Cd, Mn, and Al) has been noted for many plant species [33,34]. However, several reports indicate that lignin biosynthesis is not a universal plant response to TMs. For example, Cd and Pb at the same concentrations had opposite effects on lignin accumulation in the roots of soybean and lupine plants [35]. Environmental pollution led to an increase in the lignin content in Scots pine wood but did not affect the lignin content in *Quercus robur* L. wood [36].

The lignification of Scots pine roots increases in response to Al or Cd [33]. These results indicate that increased lignification might be an adaptive strategy of Scots pine to TMs. In this regard, the observed decrease in the content of phenolic compounds in the roots of Zn-stressed plants could be caused by the intensive synthesis of lignin. However, we did not find a stress-dependent increase in the lignin content in the organs of the seedlings (Figure 5). By contrast, the lignin content decreased both in the roots and in the needles of the Zn-stressed seedlings against the background of an increased content of phenolic compounds in the needles. This once again highlights the specificity of the plant’s protective response to the toxicity of a particular TM.

In herbaceous plants, excess Zn can cause a significant increase in lignin content. However, there is little information about the lignin content in woody plants in response to Zn toxicity. Excessive Zn increases lignification in mangrove roots, which exhibit a thick exodermis with high lignification [33]. However, until now, there has been no information on the lignin content in Scots pine organs in response to Zn toxicity [34]. Thus, it can be argued that Zn toxicity does not increase root lignification in Scots pine seedlings. Moreover, the removal of excess Zn from the nutrient solution did not lead to an increase in the lignin content in either the roots or needles of the recovering plants compared with that the roots or needles of the Zn-stressed plants. We believe that one of the reasons for this phenomenon is the intensification of the growth of young roots and needles of recovering plants.

Lignification is an important defense mechanism against bacterial and fungal pathogens [34]. Thus, the decreased lignin content in the organs of Zn-stressed Scots pine seedlings, as well as in the organs of recovering seedlings, may indicate increased sensitivity of these plants to pathogens [37].

## 4. Materials and Methods

### 4.1. Experimental Design

Seeds of Scots pine (*Pinus sylvestris* L.) were obtained from the Training and Experimental Forestry Enterprise of the Bryansk State Technological University of Engineering (Bryansk, Russia) [38]. The seedlings were cultivated in hydroculture with ZnSO_4_ [concentrations of 1.26 (control) and 150 µM], as described earlier [14]. The seedlings were cultivated on 4 L plastic trays in a growth chamber that provided a constant air temperature of 24 ± 2 °C and a 16 h photoperiod under fluorescent lighting (L36 W/765, JSC OSRAM, Smolensk, Russia, 130 ± 15 μmol m^−2^ s^−1^). After 46 days, the trays were randomly distributed among three experimental groups (control, 150 µM ZnSO_4_, and recovery (150 µM ZnSO_4_ transferred to control)) with 3 trays (approximately 450 seedlings at the beginning of the experiment) per group. Samples at the initial point (0) were collected after 46 days of growth. The seedling roots from the recovery variant were rinsed with distilled water before being transferred to nutrient solution supplemented with 1.26 µM ZnSO_4_ (control) [24]. The experiment progressed for 28 days after the transfer of the metal-stressed plants to the control group. Nutrient solutions were constantly aerated with air and renewed twice a week according to the experimental treatments. The plants were collected on the 1st, 3rd, 5th, 7th, 10th, 14th, 17th, 21st, and 28th days of the experiment.

### 4.2. Morphometric Parameters and Water Content

The lengths of the seedling organs (primary roots, hypocotyls, cotyledons, epicotyls), the diameter of the hypocotyls, and the distance from the tip of the primary root to the first lateral root were measured to an accuracy of 0.1 mm, and the numbers of first-order and second-order lateral roots and needles of the seedlings were counted in MapInfo Professional v. 9.5 software (Pitney Bowes Software, Stamford, CT, USA) [14].

The dry weight (DW) of the roots, hypocotyls, cotyledons, and needles was determined using an analytical balance (AB54-S, Mettler Toledo, Greifensee, Switzerland) with an accuracy of 0.1 mg after drying the samples at 70 °C to a constant weight. The water content of the seedling organs was expressed as a percentage of their fresh weight (FW) [24].

### 4.3. Determination of the Contents of Essential Elements

The roots of the seedlings were washed in a 20 mM aqueous solution of Na_2_-EDTA for 5 min to desorb metal ions from the root surface and then washed with distilled water [39]. After this, the plants were divided into organs (roots, hypocotyls, cotyledons, and needles) and dried until a constant weight was achieved. Thereafter, the samples were digested in concentrated HNO_3_ and HClO_4_ (2:1 (*v*/*v*)) [14]. The contents of zinc, magnesium, iron, manganese, and copper ions were determined by an AA-7000 atomic absorption spectrophotometer (Shimadzu, Kyoto, Japan).

### 4.4. Analysis of Low-Molecular-Weight Antioxidants

The low-molecular-weight antioxidants were extracted with 80% methanol from samples of the plant material ground in liquid nitrogen [24].

The low-molecular-weight antioxidant capacity (Trolox Equivalent Antioxidant Capacity (TEAC)) was determined spectrophotometrically with a Genesys 10 UV–Vis spectrophotometer (Thermo Fisher Scientific, Waltham, MA, USA), according to the method described by Re et al. [40] involving methanolic plant extracts with 2,2′-azino-bis [3-ethylbenzothiazoline-6-sulphonic acid] diammonium salt (ABTS) (Sigma, St. Louis, MO, USA, CAS Number 30931-67-0).

The total phenolic compounds content was determined spectrophotometrically using Folin and Ciocalteu’s phenol reagent (Sigma–Aldrich, MDL Number MFCD00132625), according to the procedure described by Singleton and Rossi [41]. The total phenolic content was expressed as gallic acid equivalents (GAE) in mg/g DW.

The total content of catechins and proanthocyanidins (PAs) was determined spectrophotometrically by reacting catechins and PAs with 1% vanillin in acidic medium [42]. The catechins and PAs content was calculated by constructing a calibration curve using (+)-catechin hydrate and was expressed as mg of (+)-catechin/g DW [24].

The PAs content was determined by reaction with butanol reagent. Fifty microliters of extracted sample was mixed with 700 μL of butanol reagent and heated at 95 °C for 45 min [43]. Total PAs were calculated by constructing a calibration curve using cyanidin chloride (phyproof^®^, PHL80022) and were expressed as cyanidin equivalents in mg/g DW [24].

### 4.5. Cell Wall Preparation and Lignin Quantification

Cell wall preparation was performed according to Lange et al. [44] with minor modifications. Samples of roots, needles, and epicotyls (approximately 200 mg) were ground in liquid nitrogen. The resulting fine powder was suspended in 1.5 mL of methanol (LiChrosolv^®^, Merck, Darmstadt, Germany, CAS 67-56-1). The mixture was vigorously stirred for 15 min and centrifuged (12,000× *g*, 5 min) in an Eppendorf 5417R centrifuge (Eppendorf, Hamburg, Germany). The pellet was consecutively treated with 1.5 mL of the following solvents, incubated in an ultrasonic bath (Sapphire 10456, Sapphire, Kazan, Russia), and mixed for 15 min on a Rotamix RM-1 (ELMI Ltd., Riga, Latvia), followed by centrifugation for 5 min as described above: (a) methanol (twice), (b) 1 M NaCl, (c) 1% (*w*/*v*) sodium dodecyl sulfate (Panreac, Barcelona, Spain, CAS 151-21-3), (d) distilled H_2_O (twice), (e) ethanol, (f) chloroform/methanol (l:l, *v*/*v*), and (g) tert-butyl methyl ether (Scharlau, Barcelona, Spain, CAS 1634-04-4). The remaining insoluble material (purified cell walls) was freeze-dried (Labconco FreeZone 2.5 L Benchtop Freeze Dry System, Kansas City, KS, USA) overnight [45].

Lignin was assayed by derivatization with thioglycolic acid (Sigma, CAS 68-11-1) [44]. Approximately 10 mg of the purified cell wall material was placed in a 1.5 mL screw-cap tube (SSIbio, Lodi, CA, USA, 2230-00) and treated with 1 mL of 2 M HCl and 0.2 mL of thioglycolic acid for 4 h at 95 °C. After cooling to room temperature, the mixture was centrifuged for 20 min at 20,000× *g*. The supernatant was removed, and the remaining pellet was washed three times with H_2_O. The pellet was suspended in 1 mL of 0.5 M NaOH and vigorously shaken overnight to extract the LTGA. Following centrifugation, as described above, the supernatant was transferred to a 2 mL centrifuge tube, and the pellet was washed with 0.5 mL of 0.5 M NaOH. The combined alkali extract was acidified with 0.3 mL of concentrated HCl, and the LTGA was allowed to precipitate at 4 °C for 4 h. The mixture was subsequently centrifuged, as described above, the supernatant was removed, and the brown pellets were dried in a Vacufuge Plus (Eppendorf, Germany). The pellet was dissolved in 1 mL of 0.5 M NaOH. The absorbance of each sample was measured spectrophotometrically at a wavelength of 280 nm. A calibration curve was obtained using alkali lignin (Sigma–Aldrich, CAS Number 8068-05-1) [45].

### 4.6. Determination of Photosynthetic Pigments

The chlorophyll *a* (Chl *a*), *b* (Chl *b*), and carotenoid (Car) contents were determined in the samples of basal and top needles triturated with 80% acetone. The absorbance of the samples was measured at wavelengths of 470, 646, and 663 nm, respectively [32]. The content of the photosynthetic pigments was determined using the Lichtenthaler formulas for 80% acetone [46].

### 4.7. Statistical Analysis

The number of biological replicates used for determining the fresh weight of the seedlings and for determining the morphometric parameters ranged from 42 to 60. Twelve to fifteen biological replicates were used to determine the fresh weight of the seedling organs. Four to seven biological replicates of hypocotyls and cotyledons and 7 replicates of roots and needles were used to determine the nutrient content, dry weight, and water content. Six biological replicates of needles and 4–8 biological replicates of other organs were used for the determination of phenolic compound and lignin contents. Five biological replicates were performed to determine the pigment content. Statistical analyses of the data were performed using SigmaPlot 12.5 (Systat Software Inc., Chicago, IL, USA) with one-way analysis of variance (ANOVA) followed by Duncan’s post hoc test. To separate the effects of variant and sampling time, 2-way ANOVA was used (*p* < 0.05) using SigmaPlot 12.5. Pairwise comparisons of the means with controls at corresponding time points were performed using the Student’s *t*-test for normally distributed data (significant differences at *p* < 0.05 denoted by asterisks (*)) or the Mann–Whitney rank sum test when the *t*-test was not applicable (significant differences at *p* < 0.05 denoted by multiplication symbols (×)) using SigmaPlot 12.5. The totals presented in the tables and figures are the arithmetic mean values ± standard errors.

## 5. Conclusions

The data obtained indicate that Scots pine seedlings, in a matter of days, are able to recover the growth of the primary root after the removal of excess Zn from the nutrient solution. However, this was not accompanied by promoted lateral root initiation and emergence from the parental roots. In contrast to the rapid recovery of primary root growth, the recovering seedlings did not increase the growth of the epicotyls throughout the experiment but accelerated the formation of new needles. This finding indicates the recovery of shoot apical meristem activity.

The recovery of root growth was accompanied by increased uptake of Mn, Mg, and Cu. Moreover, greater transport rates of Mn, Mg, and Fe into the needles of the recovering seedlings were observed. Despite the high recovery rate of Mn content in the roots, the Mn content in the needles did not reach the control level. The recovery of ion homeostasis eliminated all of the negative effects on the photosynthetic pigment content in the needles of the recovering plants. The progressive dehydration of the needles and a decrease in the Mn content were the most striking effects of Zn toxicity on the Zn-stressed plants throughout the experiment. Therefore, the removal of excess Zn from the nutrient solution was accompanied by the recovery of the water content in the needles of the seedlings. However, despite these obvious effects, the weight gain of the recovering seedlings did not increase compared with that of the Zn-stressed seedlings, possibly due to delayed Zn toxicity. The amount of Zn in the organs of the seedlings did not change throughout the experiment, with the exception of Zn leaching from the roots during the initial period of the experiment. In general, the Zn content in aboveground organs decreased only because of the growth of new organs.

The removal of excess Zn from the nutrient solution resulted in a slight increase in the content of low-molecular-weight antioxidants in the roots of the seedlings and a decrease in their content in the needles. However, the increased level of phenolic compounds in needles persisted for a long period after excess Zn was removed from the nutrient solution. Zn toxicity was accompanied by a decrease in the lignin content in the roots and a progressive decrease in the lignin content in the needles. This effect was not eliminated either in the roots or in the needles of recovering seedlings after the removal of excess Zn from the nutrient solution. This may create certain risks for Scots pine plants, increasing their susceptibility to pathogens. However, this assumption requires further confirmation in field experiments.

Thus, our data indicate that plants exposed to long-term Zn toxicity at the recovery stage exhibit rapid recovery of primary root growth, increased uptake of essential elements, and the restoration of several important physiological functions.

## Figures and Tables

**Figure 1 plants-13-02227-f001:**
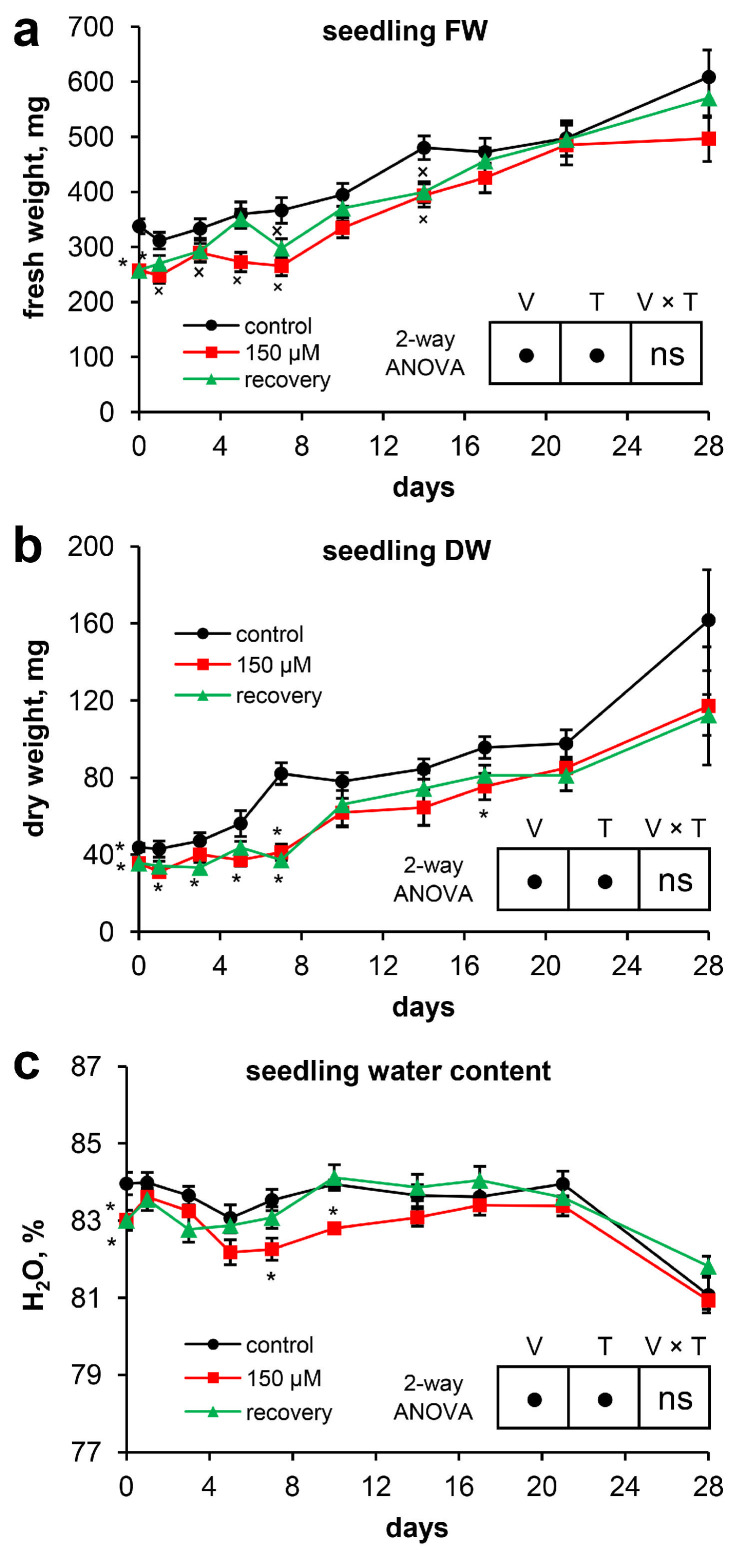
The development of Scots pine seedlings throughout the experiment: (**a**) fresh weight; (**b**) dry weight; and (**c**) water content. Pairwise comparisons of the means with controls at corresponding time points were performed using the Student’s *t*-test for normally distributed data (significant differences at *p* < 0.05 denoted by asterisks (*)) or the Mann–Whitney rank sum test when the *t*-test was not applicable (significant differences at *p* < 0.05 denoted by multiplication symbols (×)). The significance of the variant (V), sampling time (T), and variant × time (V × T) interactions were calculated using 2-way ANOVA (*p* < 0.05), with a circle (•) indicating a significant difference and “ns” indicating no significant difference.

**Figure 2 plants-13-02227-f002:**
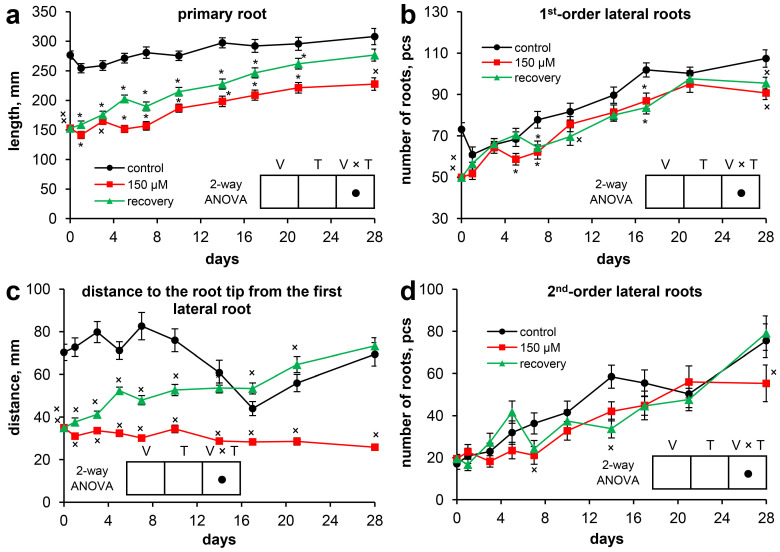
The development of the root system of Scots pine seedlings throughout the experiment: (**a**) primary root length; (**b**) number of first-order lateral roots; (**c**) distance from the tip of the primary root to the first lateral root; and (**d**) number of second-order lateral roots. Pairwise comparisons of the means with controls at corresponding time points were performed using the Student’s *t*-test for normally distributed data (significant differences at *p* < 0.05 denoted by asterisks (*)) or the Mann–Whitney rank sum test when the *t*-test was not applicable (significant differences at *p* < 0.05 denoted by multiplication symbols (×)). The significance of the variant (V), sampling time (T), and variant × time (V × T) interactions were calculated using 2-way ANOVA (*p* < 0.05), with a circle (•) indicating a significant difference.

**Figure 3 plants-13-02227-f003:**
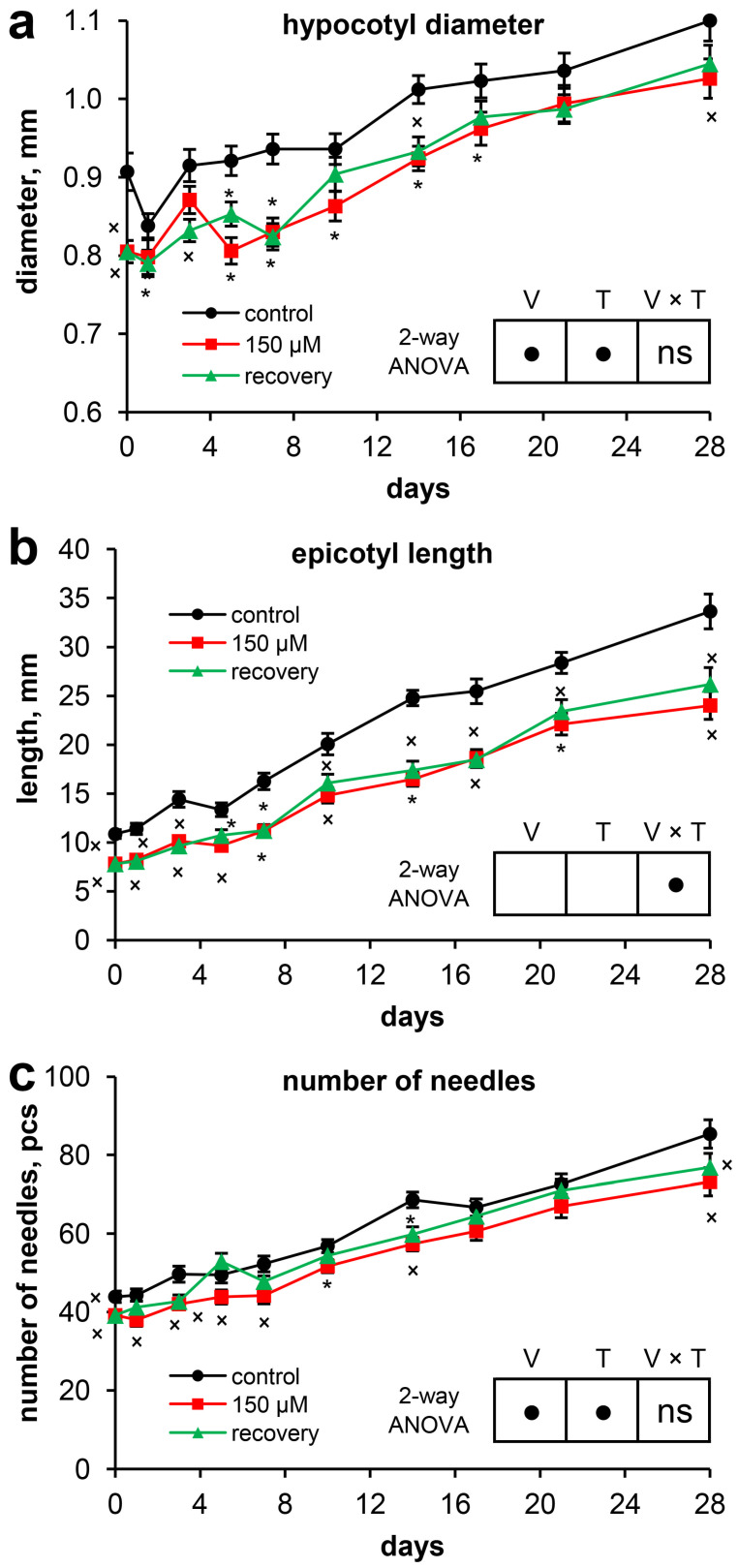
The growth of the above-ground organs of Scots pine seedlings throughout the experiment: (**a**) hypocotyl diameter; (**b**) epicotyl length; and (**c**) number of needles. Pairwise comparisons of the means with controls at corresponding time points were performed using the Student’s *t*-test for normally distributed data (significant differences at *p* < 0.05 denoted by asterisks (*)) or the Mann–Whitney rank sum test when the *t*-test was not applicable (significant differences at *p* < 0.05 denoted by multiplication symbols (×)). The significance of the variant (V), sampling time (T), and variant × time (V × T) interactions were calculated using 2-way ANOVA (*p* < 0.05), with a circle (•) indicating a significant difference and “ns” indicating no significant difference.

**Figure 4 plants-13-02227-f004:**
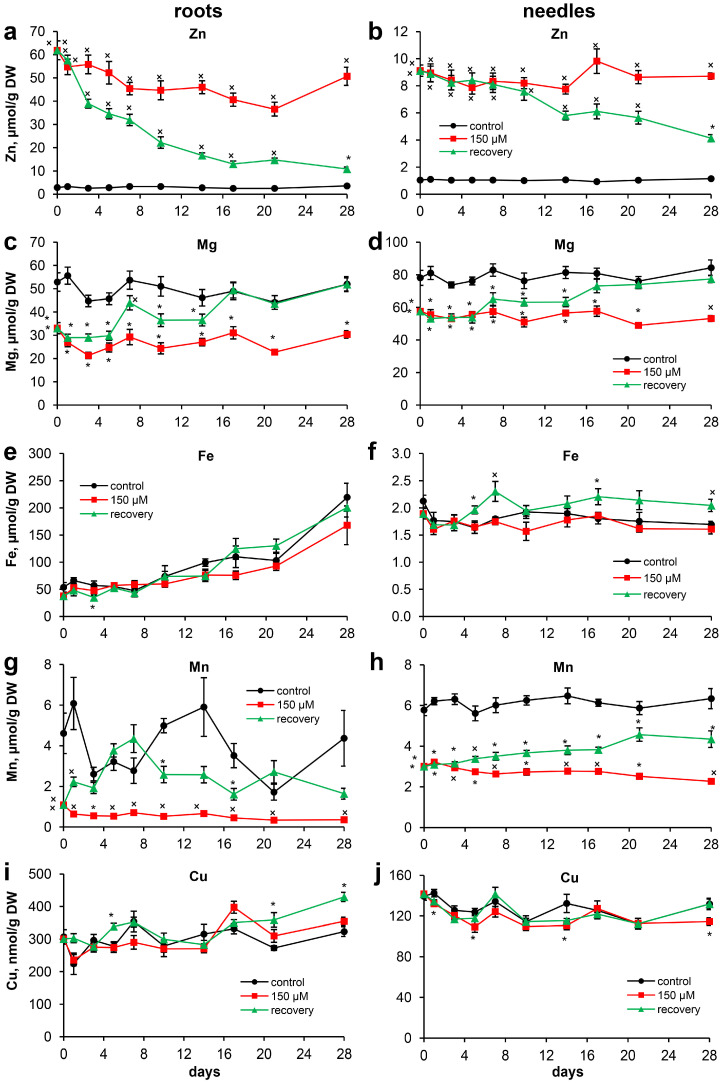
The nutrient contents: (**a**,**b**) Zn; (**c**,**d**) Mg; (**e**,**f**) Fe; (**g**,**h**) Mn; and (**i**,**j**) Cu in the roots (**a**,**c**,**e**,**g**,**i**) and needles (**b**,**d**,**f**,**h**,**j**) of Scots pine seedlings throughout the experiment. Pairwise comparisons of the means with controls at corresponding time points were performed using the Student’s *t*-test for normally distributed data (significant differences at *p* < 0.05 denoted by asterisks (*)) or the Mann–Whitney rank sum test when the *t*-test was not applicable (significant differences at *p* < 0.05 denoted by multiplication symbols (×)).

**Figure 5 plants-13-02227-f005:**
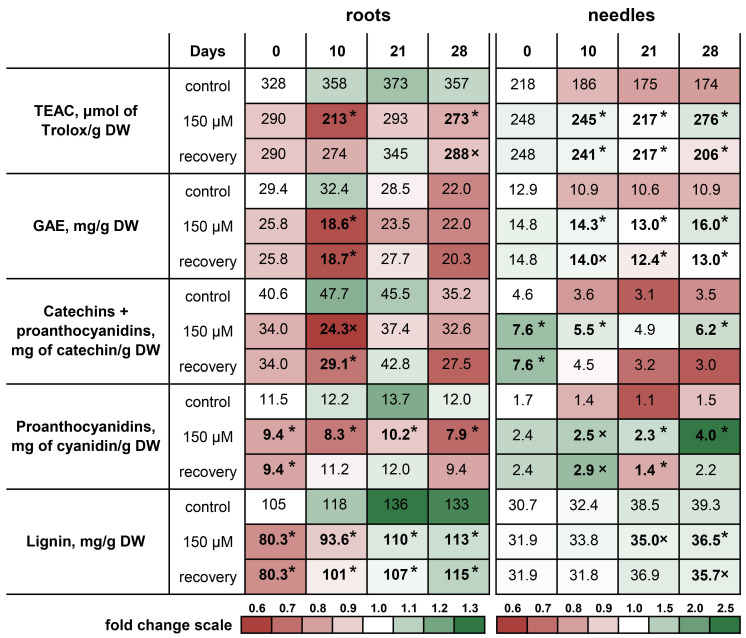
Heatmap analysis of low-molecular-weight antioxidant and lignin contents in the roots and needles of Scots pine seedlings during the experiment. The value of a given parameter in the control plants at the initial point was taken as 1.0 (white); the relative increase is indicated in green, and the relative decrease is indicated in red. Pairwise comparisons of the means with controls at corresponding time points were performed using the Student’s *t*-test for normally distributed data (significant differences at *p* < 0.05 denoted by asterisks (*)) or the Mann–Whitney rank sum test when the *t*-test was not applicable (significant differences at *p* < 0.05 denoted by multiplication symbols (×)).

**Figure 6 plants-13-02227-f006:**
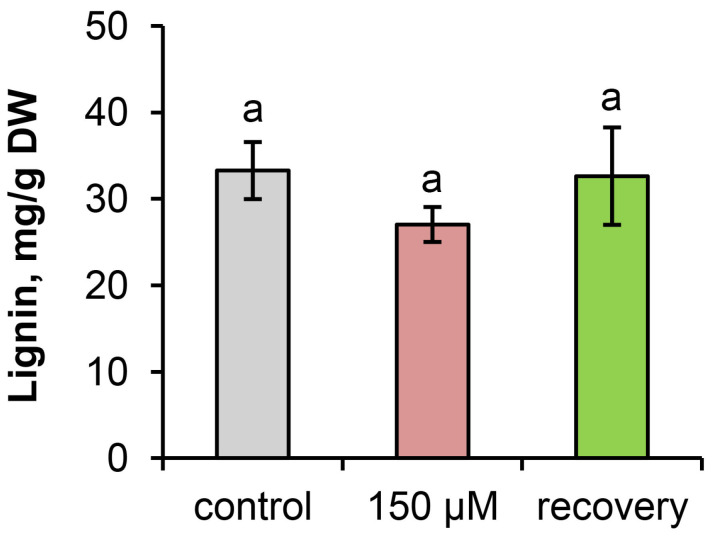
Lignin content in the epicotyls (after removing the needles) of the plants on the 28th day of the experiment. Statistical analyses of the data were performed with one-way ANOVA followed by Duncan’s post hoc test. Identical lowercase letters indicate that there are no differences between the experimental groups.

**Table 1 plants-13-02227-t001:** Growth parameters of 46-day-old Scots pine seedlings exposed to long-term excess ZnSO_4_.

Parameter	Roots		Hypocotyls		Cotyledons		Needles	
	Control	150 µM	Control	150 µM	Control	150 µM	Control	150 µM
Zinc content, µmol/g DW	2.85 ± 0.34	61.88 ± 4.05 ×	1.46 ± 0.14	12.25 ± 1.13 *	1.47 ± 0.06	9.64 ± 0.54 ×	1.04 ± 0.05	9.12 ± 0.42 ×
Fresh weight, mg	84.7 ± 4.1	61.8 ± 2.5 ×	14.9 ± 0.5	14.0 ± 0.4	21.1 ± 0.8	17.9 ± 0.7 ×	189.0 ± 12.3	138.4 ± 5.9 ×
Dry weight, mg	6.71 ± 0.44	4.90 ± 0.36 *	4.03 ± 0.08	3.59 ± 0.18	3.91 ± 0.09	4.06 ± 0.22	33.03 ± 2.48	23.04 ± 1.59 *
Water content, %	91.3 ± 0.2	91.1 ± 0.4	72.1 ± 1.9	74.3 ± 0.6	78.9 ± 1.0	78.0 ± 0.9	82.0 ± 0.2	81.1 ± 0.2 *
Length, mm	276.8 ± 7.0	152.9 ± 3.5 ×	33.7 ± 0.6	31.6 ± 0.5 *	26.6 ± 0.4	24.4 ± 0.4 ×	ND	ND
Diameter, mm	ND	ND	0.91 ± 0.02	0.81 ± 0.01 ×	ND	ND	ND	ND

Pairwise comparisons of the means with controls were performed using the Student’s *t*-test for normally distributed data (significant differences at *p* < 0.05 denoted by an asterisk (*)) or the Mann–Whitney rank sum test when the *t*-test was not applicable (significant differences at *p* < 0.05 denoted by a multiplication symbol (×)). The data on root length indicate the length of primary root. ND—no data.

**Table 2 plants-13-02227-t002:** The dynamics of fresh and dry weights and water content in the organs of Scots pine seedlings during the experiment.

Variant	Initial Point	Day of the Experiment								
		1st	3rd	5th	7th	10th	14th	17th	21st	28th
Fresh weight, mg										
Roots										
Control	84.7 ± 4.1	66.6 ± 5.4	73.3 ± 5.4	85.3 ± 9.6	84.5 ± 10.3	81.9 ± 7.1	110.1 ± 8.1	112.6 ± 11.1	132.6 ± 13.7	146.1 ± 17.1
150 µM Zn	61.8 ± 2.5 ×	58.0 ± 4.6	69.0 ± 4.0	59.4 ± 4.2	58.3 ± 6.2	80.0 ± 7.4	102.9 ± 7.5	114.3 ± 12.2	147.3 ± 18.3	120.6 ± 21.6
Recovery	61.8 ± 2.5 ×	61.1 ± 5.0	71.8 ± 9.2	80.0 ± 6.9	68.8 ± 6.5	93.0 ± 11.6	106.3 ± 8.9	123.5 ± 10.5	147.6 ± 15.4	136.6 ± 14.2
Hypocotyls										
Control	14.9 ± 0.5	14.9 ± 0.6	17.0 ± 1.0	17.1 ± 1.0	18.4 ± 1.3	19.8 ± 1.2	21.9 ± 0.9	23.0 ± 1.5	24.1 ± 1.4	32.7 ± 3.5
150 µM Zn	14.0 ± 0.4	12.6 ± 0.6 *	15.5 ± 0.9	13.4 ± 0.5 ×	14.4 ± 1.0 ×	17.1 ± 1.1	19.2 ± 0.9	20.7 ± 1.6	24.6 ± 2.0	29.3 ± 3.7
Recovery	14.0 ± 0.4	14.8 ± 0.9	15.3 ± 0.9	18.1 ± 0.9	14.7 ± 0.7	19.1 ± 1.3	20.5 ± 1.2	22.3 ± 1.5	23.8 ± 1.6	31.6 ± 2.4
Cotyledons										
Control	21.1 ± 0.8	21.3 ± 0.8	21.2 ± 0.7	19.9 ± 1.1	22.4 ± 1.2	21.2 ± 1.0	22.2 ± 0.8	22.0 ± 0.8	22.1 ± 0.9	21.7 ± 0.9
150 µM Zn	17.9 ± 0.7 ×	18.1 ± 0.9 ×	18.3 ± 0.9 ×	15.6 ± 0.7*	17.0 ± 1.0×	17.8 ± 1.1 *	18.0 ± 0.8 ×	18.8 ± 1.1 ×	17.7 ± 1.1 *	16.0 ± 1.0 *
Recovery	17.9 ± 0.7 ×	19.2 ± 0.7	17.8 ± 0.7 *	19.3 ± 0.8	17.4 ± 0.7×	20.7 ± 1.5	18.6 ± 0.6 ×	18.6 ± 0.9 *	18.5 ± 0.8 *	16.4 ± 0.9 ×
Needles										
Control	189.0 ± 12.3	182.0 ± 12.6	202.7 ± 13.7	214.4 ± 18.0	236.6 ± 21.2	254.1 ± 18.6	312.4 ± 18.1	303.0 ± 20.3	321.8 ± 21.4	468.9 ± 58.1
150 µM Zn	138.4 ± 5.9 ×	131.9 ± 9.3 *	157.0 ± 8.7 ×	154.8 ± 9.2 *	151.7 ± 13.8 ×	197.3 ± 15.7	232.9 ± 18.1 *	246.4 ± 21.9	282.6 ± 27.5	339.0 ± 56.2 *
Recovery	138.4 ± 5.9 ×	151.0 ± 11.9	158.8 ± 13.0 ×	199.0 ± 13.2	170.1 ± 12.6 ×	218.4 ± 17.7	234.2 ± 15.6 ×	275.7 ± 20.1	302.8 ± 25.4	381.7 ± 38.4
Dry weight, mg										
Roots										
Control	6.71 ± 0.44	5.44 ± 0.39	5.69 ± 0.55	8.01 ± 1.07	10.20 ± 0.74	9.30 ± 0.36	11.18 ± 0.94	13.16 ± 1.34	13.10 ± 1.07	20.66 ± 2.78
150 µM Zn	4.90 ± 0.36 *	4.40 ± 0.35	5.60 ± 0.49	5.06 ± 0.33 *	4.87 ± 0.40 *	7.68 ± 0.60 *	8.49 ± 0.97	10.66 ± 0.74	12.90 ± 1.74	15.71 ± 3.49
Recovery	4.90 ± 0.36 *	4.68 ± 0.48	5.53 ± 0.95	6.33 ± 0.45	5.27 ± 0.64 *	8.07 ± 1.21	9.66 ± 0.72	12.24 ± 0.92	10.46 ± 1.07	16.98 ± 2.02
Hypocotyls										
Control	4.03 ± 0.08	3.82 ± 0.23	4.55 ± 0.38	4.89 ± 0.42	6.61 ± 0.33	6.37 ± 0.30	6.81 ± 0.25	7.61 ± 0.55	7.53 ± 0.56	12.54 ± 2.01
150 µM Zn	3.59 ± 0.18	3.06 ± 0.28	3.84 ± 0.07	3.73 ± 0.26	4.32 ± 0.48 *	5.27 ± 0.61	5.81 ± 0.46	6.97 ± 0.63	7.32 ± 0.93	8.99 ± 1.98
Recovery	3.59 ± 0.18	3.26 ± 0.15	3.58 ± 0.20	4.45 ± 0.18	3.66 ± 0.17 ×	5.83 ± 0.86	6.74 ± 0.67	6.38 ± 0.45	6.84 ± 0.38	9.41 ± 0.84
Cotyledons										
Control	3.91 ± 0.09	3.87 ± 0.21	3.85 ± 0.14	4.24 ± 0.39	5.09 ± 0.27	4.47 ± 0.30	4.29 ± 0.42	4.30 ± 0.11	4.64 ± 0.23	4.89 ± 0.39
150 µM Zn	4.06 ± 0.22	3.44 ± 0.31	3.64 ± 0.18	3.13 ± 0.24 *	3.65 ± 0.24 *	3.99 ± 0.21	3.51 ± 0.07 ×	4.05 ± 0.27	3.74 ± 0.40	4.33 ± 0.29
Recovery	4.06 ± 0.22	3.61 ± 0.26	3.44 ± 0.31	3.58 ± 0.14	3.36 ± 0.30 *	3.99 ± 0.49	4.12 ± 0.19	3.95 ± 0.24	3.67 ± 0.12 *	4.19 ± 0.30
Needles										
Control	33.0 ± 2.48	31.2 ± 3.13	37.0 ± 4.74	43.3 ± 5.87	58.9 ± 4.06	59.2 ± 2.99	64.7 ± 3.90	72.2 ± 3.60	73.6 ± 4.79	123.6 ± 21.3
150 µM Zn	23.0 ± 1.59 *	23.8 ± 2.65	30.8 ± 2.26	29.0 ± 2.79 *	33.0 ± 4.00 *	41.8 ± 4.66 *	46.7 ± 6.54 *	52.7 ± 4.67 *	58.9 ± 7.74	81.2 ± 21.6
Recovery	23.0 ± 1.59 *	29.0 ± 3.43	28.3 ± 3.85	33.2 ± 2.70	28.6 ± 3.15 *	45.3 ± 5.88	51.0 ± 4.85 *	57.0 ± 4.48 *	60.1 ± 6.64	77.9 ± 8.35
Water content, %										
Roots										
Control	91.3 ± 0.2	91.0 ± 0.5	91.5 ± 0.3	90.8 ± 0.4	91.5 ± 0.4	91.6 ± 0.2	91.5 ± 0.4	91.5 ± 0.3	91.9 ± 0.4	90.6 ± 0.3
150 µM Zn	91.1 ± 0.4	91.3 ± 0.3	91.5 ± 0.4	90.4 ± 0.6	90.4 ± 0.2×	91.1 ± 0.3	91.8 ± 0.3	91.8 ± 0.4	92.0 ± 0.4	90.3 ± 0.4
Recovery	91.1 ± 0.4	90.6 ± 0.6	91.1 ± 0.3	90.2 ± 0.4	91.3 ± 0.3	92.4 ± 0.5	92.2 ± 0.4	92.0 ± 0.2	91.9 ± 0.5	90.8 ± 0.4
Hypocotyls										
Control	72.1 ± 1.9	73.6 ± 0.7	71.1 ± 1.9	73.3 ± 0.7	72.7 ± 0.6	73.8 ± 1.2	70.4 ± 1.3	73.9 ± 0.9	72.2 ± 1.1	72.2 ± 0.8
150 µM Zn	74.3 ± 0.6	73.3 ± 1.2	73.2 ± 1.3	69.4 ± 0.8×	70.6 ± 1.2	71.7 ± 1.7	69.7 ± 1.2	72.0 ± 1.8	71.9 ± 0.8	73.3 ± 1.0
Recovery	74.3 ± 0.6	73.7 ± 1.2	72.7 ± 1.3	72.5 ± 1.3	71.2 ± 1.7	72.2 ± 1.5	73.0 ± 0.9	73.4 ± 1.8	72.5 ± 1.1	71.5 ± 0.7
Cotyledons										
Control	78.9 ± 1.0	80.4 ± 0.8	80.0 ± 0.8	78.4 ± 0.8	80.1 ± 1.1	80.2 ± 0.8	81.6 ± 0.8	81.5 ± 0.6	81.2 ± 0.6	80.4 ± 1.0
150 µM Zn	78.0 ± 0.9	79.1 ± 0.7	77.6 ± 0.6	77.9 ± 0.8	75.9 ± 1.0 *	79.7 ± 1.3	78.2 ± 1.4	78.0 ± 1.7	79.6 ± 1.3	76.4 ± 1.4 *
Recovery	78.0 ± 0.9	80.8 ± 0.4	79.6 ± 1.2	79.5 ± 0.4	79.5 ± 0.7	81.8 ± 1.1	79.6 ± 0.8	79.6 ± 2.3	79.8 ± 1.7	78.9 ± 0.6
Needles										
Control	82.0 ± 0.2	82.5 ± 0.2	82.3 ± 0.1	81.3 ± 0.2	81.6 ± 0.2	82.0 ± 0.2	81.7 ± 0.2	81.1 ± 0.2	81.7 ± 0.2	80.4 ± 0.2
150 µM Zn	81.1 ± 0.2 *	81.8 ± 0.2 *	81.0 ± 0.1 *	80.5 ± 0.3 *	80.7 ± 0.2 *	80.7 ± 0.1 *	80.7 ± 0.2 *	80.5 ± 0.2 *	80.1 ± 0.3 *	79.1 ± 0.2 *
Recovery	81.1 ± 0.2 *	81.5 ± 0.2 *	81.0 ± 0.2 ×	80.9 ± 0.3	81.3 ± 0.2	81.1 ± 0.2 *	80.9 ± 0.2 *	81.1 ± 0.5	81.2 ± 0.2	80.4 ± 0.2

Pairwise comparisons of the means with controls at corresponding time points were performed using the Student’s *t*-test for normally distributed data (significant differences at *p* < 0.05 denoted by asterisks (*)) or the Mann–Whitney rank sum test when the *t*-test was not applicable (significant differences at *p* < 0.05 denoted by multiplication symbols (×)).

**Table 3 plants-13-02227-t003:** Content of photosynthetic pigments.

Parameter	Variant	Day of the Experiment			
		Initial Point	10th	21st	28th
Basal needles					
Chl *a*, mg/g DW	Control	8.33 ± 0.18	8.05 ± 0.24	8.23 ± 0.32	9.43 ± 0.18
	150 µM Zn	7.72 ± 0.12 *	7.80 ± 0.16	7.72 ± 0.25	8.07 ± 0.14 *
	Recovery	7.72 ± 0.12 *	7.71 ± 0.23	7.66 ± 0.28	8.93 ± 0.23
Chl *b*, mg/g DW	Control	3.36 ± 0.07	3.17 ± 0.12	3.30 ± 0.12	3.97 ± 0.11
	150 µM Zn	3.09 ± 0.05 *	3.06 ± 0.08	3.19 ± 0.04	3.32 ± 0.05 ×
	Recovery	3.09 ± 0.05 *	3.00 ± 0.08	3.07 ± 0.13	3.70 ± 0.08
Carotenoids, mg/g DW	Control	1.31 ± 0.03	1.28 ± 0.03	1.29 ± 0.05	1.44 ± 0.03
	150 µM Zn	1.22 ± 0.02 ×	1.24 ± 0.03	1.18 ± 0.08	1.25 ± 0.03 *
	Recovery	1.22 ± 0.02 ×	1.23 ± 0.04	1.21 ± 0.04	1.39 ± 0.05
Chl *a*/Chl *b*	Control	2.48 ± 0.01	2.54 ± 0.03	2.49 ± 0.01	2.38 ± 0.02
	150 µM Zn	2.50 ± 0.01	2.55 ± 0.03	2.42 ± 0.08	2.43 ± 0.03
	Recovery	2.50 ± 0.01	2.57 ± 0.02	2.49 ± 0.02	2.41 ± 0.03
Car/Chls	Control	0.112 ± 0.000	0.114 ± 0.002	0.112 ± 0.001	0.107 ± 0.001
	150 µM Zn	0.113 ± 0.001	0.114 ± 0.001	0.107 ± 0.006	0.109 ± 0.001
	Recovery	0.113 ± 0.001	0.115 ± 0.002	0.113 ± 0.001	0.110 ± 0.002
Top needles					
Chl *a*, mg/g DW	Control	6.19 ± 0.17	5.80 ± 0.19	6.45 ± 0.34	6.14 ± 0.42
	150 µM Zn	5.66 ± 0.13	5.80 ± 0.19	5.49 ± 0.19 *	5.40 ± 0.09
	Recovery	5.66 ± 0.13	5.90 ± 0.78	5.81 ± 0.15	6.40 ± 0.24
Chl *b*, mg/g DW	Control	2.53 ± 0.07	2.34 ± 0.12	2.64 ± 0.13	2.63 ± 0.20
	150 µM Zn	2.31 ± 0.06	2.32 ± 0.10	2.28 ± 0.07 *	2.27 ± 0.05
	Recovery	2.31 ± 0.06	2.34 ± 0.31	2.39 ± 0.07	2.71 ± 0.12
Carotenoids, mg/g DW	Control	0.99 ± 0.02	0.94 ± 0.02	1.02 ± 0.04	0.97 ± 0.06
	150 µM Zn	0.92 ± 0.02 *	0.93 ± 0.03	0.88 ± 0.04 *	0.87 ± 0.02
	Recovery	0.92 ± 0.02	0.98 ± 0.14	0.92 ± 0.03	1.00 ± 0.03
Chl *a*/Chl *b*	Control	2.45 ± 0.03	2.48 ± 0.04	2.45 ± 0.03	2.35 ± 0.03
	150 µM Zn	2.46 ± 0.01	2.51 ± 0.03	2.41 ± 0.03	2.38 ± 0.02
	Recovery	2.46 ± 0.01	2.53 ± 0.03	2.43 ± 0.02	2.36 ± 0.03
Car/Chls	Control	0.115 ± 0.002	0.115 ± 0.003	0.112 ± 0.002	0.110 ± 0.001
	150 µM Zn	0.116 ± 0.001	0.115 ± 0.002	0.113 ± 0.001	0.113 ± 0.002
	Recovery	0.116 ± 0.001	0.119 ± 0.001	0.113 ± 0.001	0.110 ± 0.001

Pairwise comparisons of the means with controls at corresponding time points were performed using the Student’s *t*-test for normally distributed data (significant differences at *p* < 0.05 denoted by asterisks (*)) or the Mann–Whitney rank sum test when the *t*-test was not applicable (significant differences at *p* < 0.05 denoted by multiplication symbols (×)).

## Data Availability

The datasets generated and/or analyzed during the current study are available from the corresponding author upon reasonable request.

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
