# Peer review of "Recovery of Scots Pine Seedlings from Long-Term Zinc Toxicity"

_plants, 2024, doi:10.3390/plants13162227_

Round 1

Reviewer 1 Report

Comments and Suggestions for Authors

General comments

The paper is interesting, the results are well presented, however there are some parts that need to be improved, especially the discussion part. 

 Specific comments

Introduction

More international references on the subject, of which there are many, are lacking.

Materials and methods

Lines 748-749:   Why no data were taken between days 21 and 28 on the dynamics of fresh and dry weights and water content in the organs of Scots pine seedlings during the experiment as shown in table 2?

 Discussion

They should discuss further. The discussion needs to be improved with references to international publications. Some references should be noted:     

Accumulation of heavy metals and antioxidant responses in Pinus sylvestris L. needles in polluted and non-polluted sites. Kandziora-Ciupa, M., CiepaÅ‚, R., Nadgórska-Socha, A., Barczyk, G.

Chemical composition and biochemical changes in needles of Scots pine (Pinus sylvestris L.) stands at different stages of decline in Bulgaria. Tzvetkova, N., Hadjiivanova, Ch.

Author Response

Comments 1: The paper is interesting, the results are well presented, however there are some parts that need to be improved, especially the discussion part.

Response 1: We are grateful to the reviewer for evaluating our manuscript and providing comments. The manuscript has been revised in accordance with the reviewer’s suggestions.

Comments 2: Introduction. More international references on the subject, of which there are many, are lacking.

Response 2: In accordance with the reviewer's recommendations, citations of 6 additional references have been added to the revised version of the manuscript, namely:

            Kosolapov, V.M.; Cherniavskih, V.I.; Dumacheva, E.V.; Sajfutdinova, L.D.; Zavalin, A.A.; Glinushkin, A.P.; Kosolapova, V.G.; Kartabaeva, B.B.; Zamulina, I.V.; Kalinitchenko, V.P.; et al. Scots Pine (Pinus Sylvestris L.) Ecotypes Response to Accumulation of Heavy Metals during Reforestation on Chalk Outcrops. Forests 2023, 14, 1492, doi:10.3390/f14071492.

            Rahmonov, O.; Sobala, M.; Åšrodek, D.; Karkosz, D.; Pytel, S.; Rahmonov, M. The Spatial Distribution of Potentially Toxic Elements in the Mountain Forest Topsoils (the Silesian Beskids, Southern Poland). Sci Rep 2024, 14, 338, doi:10.1038/s41598-023-50817-7.

            ÅšwiÄ…tek, B.; Kraj, W.; Pietrzykowski, M. Adaptation of Betula Pendula Roth., Pinus Sylvestris L., and Larix Decidua Mill. to Environmental Stress Caused by Tailings Waste Highly Contaminated by Trace Elements. Environ Monit Assess 2024, 196, 52, doi:10.1007/s10661-023-12134-4.

            Spasić, M.; Vacek, O.; Vejvodová, K.; Tejnecký, V.; Vokurková, P.; Križová, P.; Polák, F.; Vašát, R.; Borůvka, L.; Drábek, O. Which Trees Form the Best Soil? Reclaimed Mine Soil Properties under 22 Tree Species: 50 Years Later—Assessment of Physical and Chemical Properties. Eur J Forest Res 2024, 143, 561–579, doi:10.1007/s10342-023-01637-x.

            Tzvetkova, N.; Hadjiivanova, Ch. Chemical Composition and Biochemical Changes in Needles of Scots Pine (Pinus Sylvestris L.) Stands at Different Stages of Decline in Bulgaria. Trees 2006, 20, 405–409, doi:10.1007/s00468-006-0052-8.

            Kandziora-Ciupa, M.; CiepaÅ‚, R.; Nadgórska-Socha, A.; Barczyk, G. Accumulation of Heavy Metals and Antioxidant Responses in Pinus Sylvestris L. Needles in Polluted and Non-Polluted Sites. Ecotoxicology 2016, 25, 970–981, doi:10.1007/s10646-016-1654-6.

Comments 3: Materials and methods. Lines 748-749:   Why no data were taken between days 21 and 28 on the dynamics of fresh and dry weights and water content in the organs of Scots pine seedlings during the experiment as shown in table 2?

Response 3: Our previous experiments revealed an increase in the Zn content in the needles of Scots pine seedlings compared with that in the control seedlings after 3 days of exposure, whereas Zn accumulation in the roots was noted throughout 14 days of exposure (Ivanov et al., 2021 Phytotoxicity of short-term exposure to excess zinc…). In this context, the design of our experiment involved a more detailed analysis of the period from day 1 to day 21 of the experiment. The time interval for collecting the last point of the experiment (28 days) was doubled on average to demonstrate differences in the physiological effects on the dynamics. We have used a similar approach previously (see for ref.: Zlobin, I.E., Ivanov, Y.V., Kartashov, A.V., Sarvin, B.A., Stavrianidi, A.N., Kreslavski, V.D. and Kuznetsov, V.V., 2019. Impact of weak water deficit on growth, photosynthetic primary processes and storage processes in pine and spruce seedlings. Photosynthesis research, 139, pp.307-323).

Comments 4: Discussion. They should discuss further. The discussion needs to be improved with references to international publications. Some references should be noted:

Accumulation of heavy metals and antioxidant responses in Pinus sylvestris L. needles in polluted and non-polluted sites. Kandziora-Ciupa, M., CiepaÅ‚, R., Nadgórska-Socha, A., Barczyk, G.

Chemical composition and biochemical changes in needles of Scots pine (Pinus sylvestris L.) stands at different stages of decline in Bulgaria. Tzvetkova, N., Hadjiivanova, Ch.

Response 4: The Discussion section has been revised in accordance with the Reviewer's recommendations.

Reviewer 2 Report

Comments and Suggestions for Authors

Dear authors,

Overall, the manuscript is well written, following the Instruction for authors, but still needs some correections to be made as follows:

1) Line 70: HMs or TMs?

2) Lines 87-92: The aim is not well fomulated. There is a lack of scientific hypothesis.

3) Lines 653-654: How do know that Mg deficiency led to a decrease of photosynthetic pigments without analyze its content?

4) Section 4.1.: Methodology of the experiment lacks any references.

5) Conclusion sections should not include references (lines 874-889). This text will be more suitable as an introduction or discussion. Conclusions should highlight your own findings and their importance for the fundamental science and/or practice but that is missing into the manuscript.

As a synopsis of the review, I could say that a lot of work has been made for both the experimental and analyses. Nevertheless, the results obtained need future validation in a real environment in order to be usefull for the phytoremediaton and restoration of Zn-contaminated areas.

Author Response

Comments 1: Dear authors, Overall, the manuscript is well written, following the Instruction for authors, but still needs some correections to be made as follows:

Response 1: We are grateful to the reviewer for the positive evaluation of our manuscript and the comments. The manuscript was revised in accordance with the reviewer's suggestions.

Comments 2: 1) Line 70: HMs or TMs?

Response 2: In general, the terms “heavy metals” and “toxic metals” are used to refer to the same elements. In recent years, the term “toxic heavy metal(loid)s (TMs)” has become more common, and we have used this term in this manuscript. To avoid ambiguity, the use of the abbreviation HMs has been excluded from the manuscript in accordance with your comments.

Comments 3: 2) Lines 87-92: The aim is not well fomulated. There is a lack of scientific hypothesis.

Response 3: We have reformulated the research objective in the revised version of the manuscript. We hope that in its current form it more fully reveals scientific hypotheses.

Comments 4: 3) Lines 653-654: How do know that Mg deficiency led to a decrease of photosynthetic pigments without analyze its content?

Response 4: On the basis of previous experiments with Mn-deficient Scots pine seedlings (Ivanov et al., 2022 Manganese Deficiency Suppresses Growth and Photosynthetic…), we found that even a more severe manganese deficiency in needles than in the current experiment does not lead to a decrease in the content of photosynthetic pigments. The iron content (Figure 4f) in the needles of the Zn-stressed seedlings did not differ from that in the control plants throughout the experiment. Thus, we believe that the reduced magnesium content in needles (Figure 4d), which is involved in chlorophyll formation (Tripathy and Pattanayak 2012) is the most likely cause of the reduced content of photosynthetic pigments in Zn-stressed plants.

Comments 5: 4) Section 4.1.: Methodology of the experiment lacks any references.

Response 5: The necessary references have been added to the description in section 4.1.

Comments 6: 5) Conclusion sections should not include references (lines 874-889). This text will be more suitable as an introduction or discussion. Conclusions should highlight your own findings and their importance for the fundamental science and/or practice but that is missing into the manuscript.

Response 6: In accordance with the reviewer's recommendations, we have removed the last paragraph of the Conclusion section. Thus, there are no references in the Conclusion section anymore.

Comments 7: As a synopsis of the review, I could say that a lot of work has been made for both the experimental and analyses. Nevertheless, the results obtained need future validation in a real environment in order to be usefull for the phytoremediaton and restoration of Zn-contaminated areas.

Response 7: We fully agree with the reviewer's opinion. The experimental system used allowed us to identify certain patterns in the physiological response of plants that are extremely difficult to detect under natural conditions, such as the growth and development of the root system. However, the results obtained cannot be extrapolated to real environments without additional research.

Reviewer 3 Report

Comments and Suggestions for Authors

The manuscript is well-written and presents a valuable study on the recovery of Scots pine seedlings from long-term zinc toxicity With minor revisions, it will be suitable for publication.

Minor Comments:

The introduction offers a solid overview of soil contamination and plant recovery. It could be improved by detailing Scots pine’s sensitivity to zinc and addressing current knowledge gaps regarding its recovery from long-term zinc exposure.

The methods section is well-written but could benefit from more details on statistical analyses, including the software used and any data transformations.

The discussion effectively interprets results and situates them within existing research. To enhance it, the authors can explore broader implications for Scots pine resilience and forest ecosystems, and expand on how decreased lignin content affects pathogen susceptibility and ecological impacts.

Comments on the Quality of English Language

Minor grammatical and typographical errors could be corrected.

Author Response

Comments 1: The manuscript is well-written and presents a valuable study on the recovery of Scots pine seedlings from long-term zinc toxicity With minor revisions, it will be suitable for publication.

Response 1: We are grateful to the reviewer for the positive evaluation of our manuscript and the comments. The manuscript was revised in accordance with the reviewer's suggestions and has received language editing from American Journal Experts.

Comments 2: The introduction offers a solid overview of soil contamination and plant recovery. It could be improved by detailing Scots pine’s sensitivity to zinc and addressing current knowledge gaps regarding its recovery from long-term zinc exposure.

Response 2: We have added citations of 6 additional references concerning the tolerance of Scots pine to toxic metals, as well as the possibility of the use of Scots pines for phytostabilization at contaminated sites. Data on the sensitivity of Scots pine to zinc were published in our previous manuscripts, which are cited in the text. We would like to draw the reviewer's attention to the fact that there are no data on the recovery of Scots pine from the toxic effects of zinc. In general, the physiological features of the recovery of woody plants from the toxic effects of metals have rarely been studied. Moreover, most studies of the toxic effects of metals on Scots pine usually address combinations of different pollutants. This significantly limits the possibility of direct comparison and interpretation of physiological effects.

Comments 3: The methods section is well-written but could benefit from more details on statistical analyses, including the software used and any data transformations.

Response 3: Statistical analyses of the data were performed using SigmaPlot 12.5 (Systat Software Inc., USA).

Comments 4: The discussion effectively interprets results and situates them within existing research. To enhance it, the authors can explore broader implications for Scots pine resilience and forest ecosystems, and expand on how decreased lignin content affects pathogen susceptibility and ecological impacts.

Response 4: The last paragraph of the Discussion section, concerning lignification as an important defense mechanism against bacterial and fungal pathogens, is, in our opinion, a logical assumption of the consequences of the decrease in lignin content in the organs of Zn-stressed Scots pine seedlings. However, additional studies, including those currently underway in our laboratory, are needed to verify and prove this hypothesis. Therefore, in this manuscript, we do not consider it possible to go into greater depth in the discussion of this aspect.

Comments 5: Comments on the Quality of English Language. Minor grammatical and typographical errors could be corrected.

Response 5: The revised version of the manuscript has received language editing from American Journal Experts.

Round 2

Reviewer 1 Report

Comments and Suggestions for Authors

The changes proposed have been done 

Reviewer 2 Report

Comments and Suggestions for Authors

Dear authors,

I think that the revised version of the manuscript is significantly improved and can be accepted for publication.